# Evaluation of the Ecological Status of Wetlands of International Importance in China

Xinsheng Zhu [1], Yongfeng Yang [1], Jun Yuan [1] and Ziru Niu [2,3,*]

[1]   Academy of Inventory and Planning, National Forestry and Grassland Administration, Beijing 100013, China
[2]   Shanxi Provincial Land Engineering Construction Group Co., Ltd., Xian 710075, China
[3]   Institute of Land Engineering and Technology, Shanxi Provincial Land Engineering Construction Group Co., Ltd., Xian 710075, China
*    Correspondence: niuziru12345@163.com

**Abstract:** An international assessment of wetlands is important for quantifying the current state of development of international wetland structure and function. At present, the management department and public lack a comprehensive understanding of the ecological status of internationally important wetlands in China. Here, using annual ecological monitoring data and hierarchical analysis, an evaluation index system was constructed with water environment, biological environment, biodiversity, and disturbance pressure as influencing factors, and an evaluation of the ecological status of 63 wetlands of international importance (WOII) in China was carried out, it is helpful to formulate technical plans for the ecological management of wetlands of international importance (WOII). The results showed that the average ecological status score of these wetlands was 0.714 ± 0.075, and there were differences in this score between different regions. The ecological status of wetlands in northeastern and northern coastal areas were mostly evaluated as 'excellent' and 'good', while some wetlands in other areas of China were evaluated as 'poor'. The type and protection level of wetlands have an important impact on their ecological status. The ecological status of wetlands according to type were ranked swamp wetlands > coastal wetlands > river wetlands > lake wetlands, while those with national level protection were ranked higher than those with provincial level protection. The ecological status of WOII in China is generally good, but is adversely affected by human activities, alien species invasion, and other factors. Hence, there needs to be a focus on improving the protection and management mechanisms for WOII, promoting improvements in the ecological status of these wetlands, and transforming ecological product value.

**Keywords:** wetlands of international importance; ecological status assessment; conservation management

## 1. Introduction

Wetlands of international importance (WOII) are wetlands of unique international significance in terms of ecology, botany, and/or zoology that meet the assessment criteria of the Convention on Wetlands of International Importance (hereafter referred to as the 'Wetlands Convention'), especially in terms of waterfowl habitat, and have been established upon application to the Wetlands Convention committee [1,2]. As the first global environmental convention, the Wetlands Convention promotes the conservation and management of WOII through a listing system, and after half a century of development, it plays an important role in protecting the ecological environment and biodiversity of wetlands and allowing them to perform their ecological functions [3]. Since China joined the Wetlands Convention in 1992, it has actively participated in wetland compliance actions. By the end of 2021, China had recognized 64 WOII (63 on the mainland and one in Hong Kong) in 11 batches, and had contributed to global wetland conservation and ecological restoration through the implementation of ecological projects, such as migratory bird network protection and invasive plant management in WOII [4].

Wetland ecological status evaluation selects the main biotic and abiotic characteristics of a given wetland, and evaluates the status of the wetland ecosystem over a certain timescale [5,6]. Previous studies have shown that regular assessments of wetland ecological status can help to understand changes in ecological status and quantify the impact of human disturbance on wetlands. Under a background of the quality and extent of global wetland areas being threatened, it is important to improve the level of wetland protection [7–9]. At present, for wetlands falling within the framework of the Wetlands Convention in South America, Oceania, Europe, and other locations, numerous studies have been carried out on ecosystem protection and landscape restoration on the basis of wetland ecological status assessment [10–13].

Since the accession of China to the Wetlands Convention and the initial establishment of a wetland protection management, investigation, and monitoring system, the evaluation of the ecological status of wetlands has developed. The evaluation index has evolved from an initial single ecological factor to a comprehensive multi-factor evaluation incorporating socio-economic factors. The evaluation index has also been combined with wetland monitoring, and various models, such as wetland ecological quality evaluation and wetland ecological value evaluation, have emerged. Meanwhile, the social and economic development of China continues to threaten wetland resources, and a contradiction between resource protection and economic development remains. Effective monitoring and evaluation of wetlands has become an important means of sustainable wetland management [14,15]. Wetlands of international importance protect the habitats of a large number of endangered migratory waterbirds and aquatic animals, and are an important component of China's wetland protection system. The ecological status of WOII is of great significance to the formulation of Chinese wetland protection and management policies. China implements a tiered management system for important wetlands, general wetlands, and a list of wetland resources. Currently, WOII are the main component of important wetlands at the national level. Hence, the ecological status of WOII reflects the overall status of China's important wetlands, to a certain extent.

To strengthen the management of WOII, China carried out two programs of ecological monitoring of such wetlands at the national level in 2009 and 2013, and has carried out annual monitoring of WOII since 2019. This has provided certain basic data for the formulation of international compliance policies for WOII. On this basis, the assessment of the ecological status of WOII in China has also resulted in studies on the ecological status and environmental quality of wetlands, and proposed targeted research methods and systems. However, there has been a focus on the evaluation of individual important wetlands or comparison of important wetlands in a given region [16,17], and there has been no overall assessment of the ecological status of important wetlands on a national scale in recent years. Conservation management departments and the general public lack a holistic and quantitative understanding of the real ecological status of WOII.

Therefore, the main questions of this paper are: (1) What is the index system suitable for the ecological quality evaluation of WOII in China; (2) What is the current ecological status of WOII in China. The main objectives of this paper are: (1) to construct an evaluation system that can be comprehensively applied to WOII in China; (2) to derive the current ecological status of WOII in China and the main influencing factors; and (3) to make reasonable suggestions for solving problems associated with WOII.

## 2. Material and Methods

### 2.1. Study Area

The study area comprised of 63 WOII (excluding Mai Po Marshes in Hong Kong and Inner Houhai Bay WOII) within the annual ecological status monitoring database of wetlands in China, distributed across 23 provinces (autonomous regions and municipalities directly under the central government). Heilongjiang Province has the largest number of WOII, with 10, followed by the Inner Mongolia Autonomous Region, Hubei Province, Guangdong Province, Yunnan Province, Tibet Autonomous Region, and Gansu Province,

with four each. According to statistical data, these 63 WOII comprise a total area of 7,327,000 hectares, within the wetland area of 3,727,500 hectares, Wetland rate 50.88%, lake wetlands and marsh wetlands are the main wetland types (Figure 1).

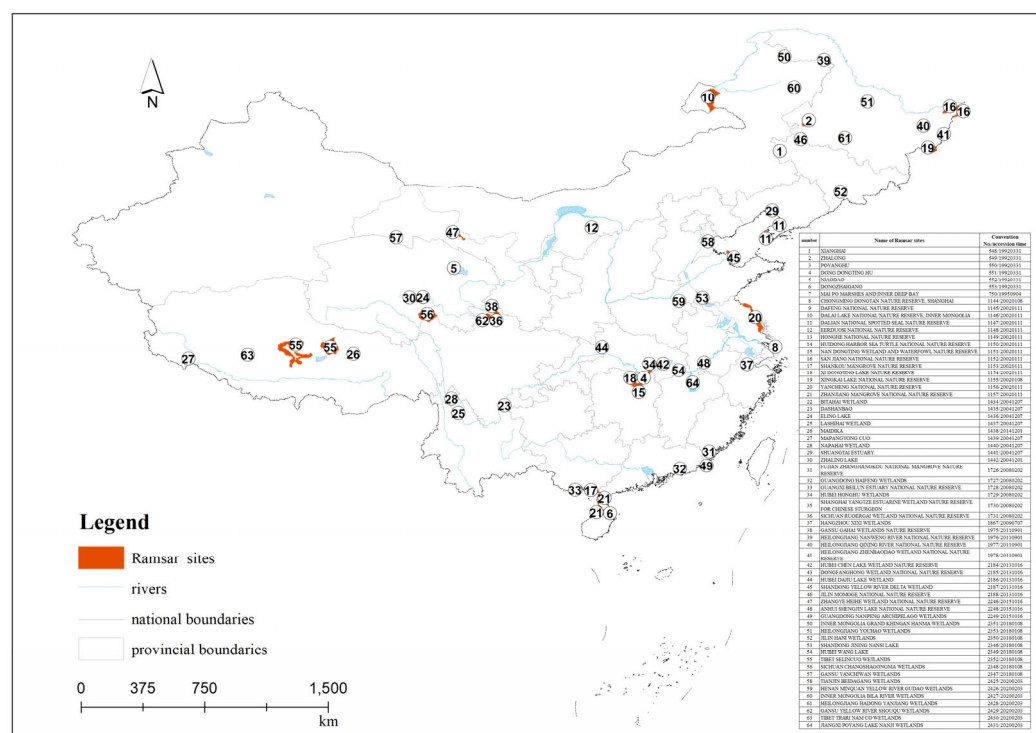

**Figure 1.** Ramsar sites distribution in China.

## 2.2. Data and Methods

Evaluation Indicators

The degree of excellence of the ecological environment of wetlands includes the quality of the water environment, habitat quality, species diversity, and any external disturbances [18,19]. In this study, we established an index system for evaluating the ecological status of WOII, and carried out an evaluation of the ecological quality of WOII in China on the basis of individual WOII with the joint participation of management departments at all levels of WOII. The indicator system was constructed by combining the 'Monitoring and Assessment of Ecological Status of Wetlands of International Importance', 'Wetland Ecological Quality Assessment Specification' (DB11/T 1503-2017), and other normative technical documents and relevant research results. It consists of three levels of indicators, including the area of wetlands, water recharge status, surface water quality, eutrophication status of water bodies, wetland plants, wetland bird distribution, and precious/endangered species. In this case, 12 specific evaluation factors, such as species and population status, invasive exotic plant status, land (water) use, and others, were designed to objectively reflect the environmental status and ecological characteristics of different types of WOII across different regions. According to the relative importance of each evaluation index and priority of wetland ecosystem protection, combined with standard classification and standard specifications requirements, the evaluation indices were graded and index weights determined. According to the identification of index weights by experts in the field, if the index has a greater impact on ecological quality, a higher weight value will be assigned.

The ecological status evaluation indicator data for WOII were obtained from the website of the Wetlands Convention (https://www.ramsar.org, accessed on 1 October 2022), combined with in-situ survey verification and data collection, among which the indicators of water environment and biodiversity were mostly taken from official reports provided

by the management departments of the protected areas where the WOII are located, or the environmental status bulletins of the locations of important wetlands. Indicators of biological environment and disturbance pressure were mostly obtained from analysis of surface coverage of remote sensing images and field surveys.

In the Table 1, the water environment assessment content mainly refers to ecological factors, such as wetland hydrological conditions and water quality, that have a greater impact on wetland evolution processes [20]. In the bioenvironmental assessment content, the proportion of bioenvironmental habitat area and vegetation coverage are the most direct indicators reflecting ecological structure and function [21,22]. The biodiversity assessment content is dominated by wetland birds [23], comprehensively calculated with reference to species richness, diversity, abundance, and evenness, with an increased weight for endangered species in the indicators [24,25]. The disturbance pressure assessment content is based on the disturbance degree model comprising of wetland biological invasion and land use change, and spatially and quantitatively studies the impact of external disturbance on wetland ecological status [26,27].

**Table 1.** The weight value and assignment of ecological status evaluation index in Ramsar sites.

| Assessment Content | Evaluation Indicators | Evaluation Factors | Wij | Assignment Criteria |
|---|---|---|---|---|
| quality of the water environment | water resource | water supply status of wetland | 0.08 | discharge of water: fully satisfied/basically satisfied/ insatiable |
| | Water quality | surface water environmental quality | 0.12 | water quality: Class II and above/Class III/Class IV/Class V and below |
| | | water eutrophication | 0.06 | eutrophication degree: poor nutrition/medium nutrition/eutrophication |
| habitat quality | Vegetation coverage | vegetation coverage density | 0.06 | proportion of vegetation area in wetland: 80%/50%/10% |
| | Plant diversity | wetland plant richness | 0.06 | richness index: 6/5.5/4.5 |
| | Habitat integrity | wetland area | 0.10 | wetland area: one hundred thousand hectares/ ten thousand hectares/one thousand hectares |
| species diversity | Species diversity | species of waterfowl | 0.07 | annual recorded type: one hundred/fifty |
| | | number of waterfowl | 0.10 | annual recorded quantity: twenty hundred thousand / one hundred thousand / twenty thousand |
| | endangered species | species of endangered species | 0.12 | national protected animal species in China: thirty/twenty/ten |
| external disturbance | alien invasive species | invasion status | 0.10 | damage level: not formed/slightly controllable/moderately out of control |
| | degree of human disturbance | wetland resource utilization | 0.07 | Proportion of human disturbance area: 10%/30%/60% |
| | | land Use intensity | 0.06 | Change in ecological patch: Increase/decrease slightly/ decrease more |

Note: The value of water quality condition index is the average result measured at several points.

According to the monitoring value of the indicator, each individual indicator is assigned in the assignment standard, and the comprehensive evaluation score (SE) of the ecological status of the WOII is obtained by the weighted calculation of the analytic hierarchy process:

$$SE = \sum(F_i * W_i)$$

where, $F_i$ is the score assigned to the $i$ index, and $W_i$ is the weight of the $i$ index.

Considering that the classification statistics of index scores can more intuitively reflect the management and protection of nature reserves, according to the SE, the ecological status of WOII were divided into 'excellent', 'good', 'moderate', and 'poor'. A score $\geq 0.800$ is 'excellent', 0.700–0.800 is 'good', 0.600–0.700 is 'moderate', and <0.600 is 'poor'. One-way analysis of variance (ANOVA) was used to test the differences in scores of different types of WOII. For cases where there were significant overall differences, the independent samples *t*-test was used to compare the differences between pairs.

## 3. Results and Analysis

### 3.1. Overall Situation

The overall score of the ecological status of WOII in China was relatively high (Table 2). The ecological status score range was 0.497–0.890, with an average of 0.714 ± 0.075. According to the rating scale, six WOII including Heilongjiang Nanweng River, Jilin Xianghai, Inner Mongolia Daxing'anling Khanma were rated as 'excellent', accounting for 9.52% of the total WOII. In this case, 35 WOII, including Heilongjiang Sanjiang, Jiangxi Poyang Lake, and Shandong Yellow River Delta WOII were rated as 'good', accounting for 55.56% of the total. Here, 16 WOII, including Qinghai Eling Lake, Hubei Shenhu, and Tibet Zarinanmucuo were rated as 'medium', accounting for 25.40% of the total. Six WOII including HangzhouXixi Wetland, Hubei Net, Guangdong Huidong port turtles resulted in 'low' ratings. These results show that the overall ecological status of the 63 WOII is good, but there are clear differences in the ecological status scores.

**Table 2.** Evaluation score distribution of Ramsar sites.

| Score | $\geq 0.800$ | $0.700 \leq SE < 0.800$ | $0.600 \leq SE < 0.700$ | $\leq 0.600$ |
|---|---|---|---|---|
| quantity/piece | 6 | 35 | 16 | 6 |
| proportion/% | 9.52 | 55.56 | 25.40 | 9.52 |

### 3.1.1. Water Environment

The current status of the water environment of Chinese WOII is good, but locally there remains a threat of pollution. The water environment had an evaluation score range of 0.104–0.237, with an average of 0.180 ± 0.038, and an overall score rate of 69.08%. The headwaters of rivers, upstream, and estuarine areas scored higher than middle and downstream areas (Figure 2a). There were also differences in the status of specific sub-indicators of WOII water environment. In terms of water resources, 85.71% of the WOII rely on natural recharge, such as precipitation and surface runoff, to meet ecological needs. The water recharge status of WOII including Zaling Lake and Eling Lake in Qinghai, Khanma in Inner Mongolia, and Nanjung River in Heilongjiang in the key state-owned forested area in northeast China was evaluated as essentially stable. However, for Shanghai Chongming East Beach, Guangxi Shankou mangrove forest, Liaoning Shuangtai estuary, and other estuarine areas of WOII, where natural recharge is insufficient, artificial recharge measures have been taken.

In terms of water quality conditions, the proportion of Chinese WOII with surface water quality above Class II (including seawater), Class III, Class IV, and Class V were 46.03%, 23.81%, 14.29%, and 15.87%, respectively, with the proportions in classes II and III being the greatest. The proportions evaluated to have nutrient degrees of poor, medium, and eutrophic were 31.75%, 47.62%, and 20.63%, respectively. The water quality of WOII in the headwaters, upper reaches, and estuaries of rivers is good, but that of WOII in the middle and lower reaches of the Yangtze River in Hubei, Sink Lake and Hunan South Dongting Lake, and in the Yunnan Napa Sea of the northwest plateau is relatively poor, and eutrophication levels can be high owing to the influence of surrounding industry and farming.

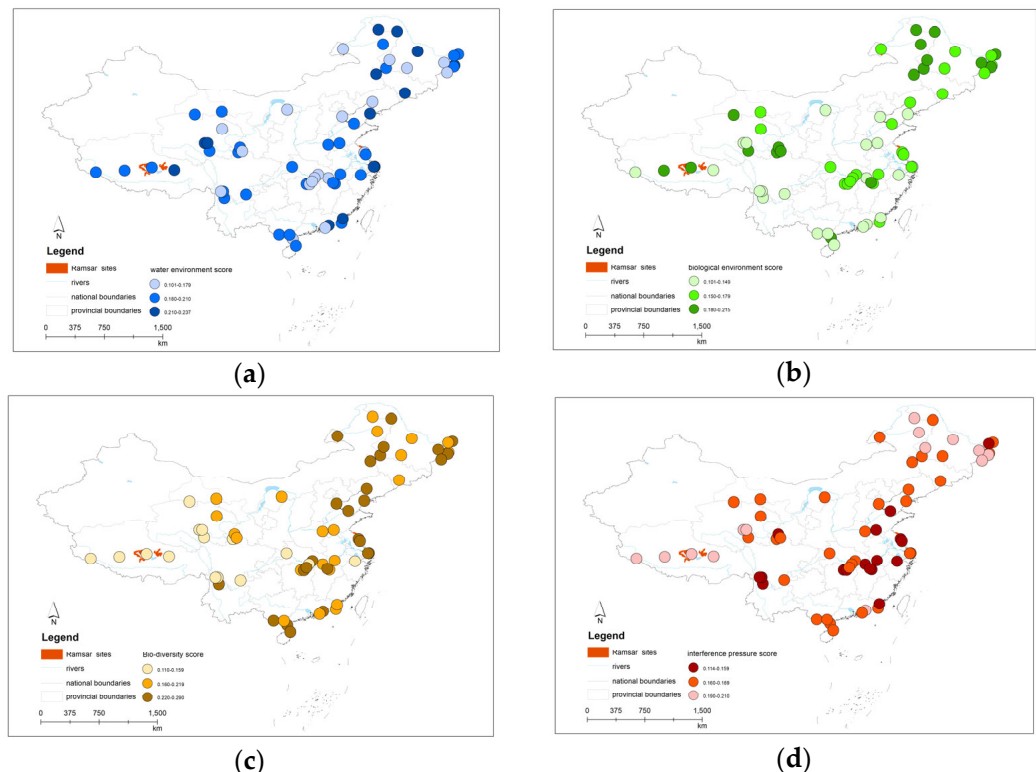

**Figure 2.** Evaluation results of the ecological status of Ramsar sites. (**a**) Results of water environment assessment, (**b**) Results of the biological environment, (**c**) Results of Bio-diversity, (**d**) Results of interference pressure.

### 3.1.2. Biological Environment

The biological environment of Chinese WOII was evaluated as being in good condition. Wetlands in the northeast and west of the country scored higher than those in other regions (Figure 2b). The biological environment scores ranged from 0.101 to 0.215, with an average of 0.159 ± 0.028, and an average score rate of 72.21%. The vegetation coverage of the 63 WOII is 1,604,800 hectares, accounting for 43.05% of the total wetland area. Vegetation coverage in wetlands such as Ruoergei in Sichuan, Nanjer River in Heilongjiang, Qixing River in Heilongjiang, and Khanma in Inner Mongolia is >90%, and the plant diversity is also rich. In contrast, in WOII estuarine areas, such as Dalian Spotted Seal in Liaoning, Huidong Harbor Sea Turtle in Guangdong, and Chinese Sturgeon in the Yangtze River Estuary, there is low vegetation cover and plant diversity. The habitat integrity of WOII is positively related to the area of the wetland. The area of WOII in the western and northeastern regions is much larger than that in the central region, and consequently the habitat integrity is relatively higher in these regions.

### 3.1.3. Biodiversity

The current status of biodiversity in Chinese WOII was not highly evaluated. The numbers and species diversity of waterbirds in most wetlands were high, in line with the characteristics of WOII as important habitats for wetland waterbirds, but there were differences in biodiversity status between different WOII. The biodiversity scores in the coastal and eastern regions were higher than those in the west (Figure 2c). The evaluated biodiversity scores ranged from 0.110 to 0.207, with an average of 0.201 ± 0.050, and a score rate of 69.20%. The overall number of species and number of key species in the WOII were analyzed. The WOII in the northeast, middle, and lower reaches of the Yellow River, middle and lower reaches of the Yangtze River, and other eastern regions are located on the East Asia–Australia migration route. In this case, the biodiversity was generally high, among which the number of waterbird species in WOII in the estuarine regions of

Shanghai Chongming Dongtan, Shandong Yellow River Delta, and Liaoning Shuangtai estuary exceeded 150. However, WOII in the eastern region, such as Dalian Spotted Seal and Yangtze River Estuary Chinese Sturgeon, are located adjacent to offshore waters, with almost no mudflat wetlands, and consequently the numbers and diversity of waterfowl species are relatively small. Among other WOII, those in western regions, such as Mabon Yongzuo in Tibet and the Napa Sea in Yunnan, as well as Xixi in Hangzhou and other WOII located in the middle of cities, have relatively few wetland species and a low degree of biodiversity.

### 3.1.4. Disturbance Pressure

Under the influence of governmental policy, WOII have been protected and the overall disturbance pressure was evaluated as low, but WOII in the middle and lower reaches of the Yangtze River are subject to relatively high disturbance pressure and scored lower than other regions (Figure 2d). The disturbance pressure score ranged from 0.121 to 0.210, with an average of 0.174 ± 0.019, and the highest score rate of 75.86%. Specific statistical results show that 41.27% of the WOII are threatened by invasive alien plants, among which coastal areas, such as the Yellow River Delta in Shandong and Chongming East Beach in Shanghai, are most seriously threatened. In this case, 65.35% of the WOII are threatened by unreasonable industrial and agricultural development and utilization activities, among which WOII in the middle and lower reaches of the Yangtze River and southwestern areas, such as Hong Lake in Hubei and the Napa Sea in Yunnan, are most affected by environmental pollution. The WOII in northeastern and western regions, such as Sanjiang in Heilongjiang and Madika in Tibet, are most affected by agriculture and animal husbandry, and estuarine regions, such as Shuangtaihekou in Liaoning and Yancheng in Jiangsu, are most affected by infrastructure construction.

### 3.2. Ecological Status of Different Types of WOII

According to the Wetlands Convention and national wetland survey classification system, Chinese WOII were classified into two categories: offshore–coastal and inland wetlands. Inland wetlands were further classified into marsh wetlands, lake wetlands, and river wetlands, according to their different characteristics [28]. When comparing the ecological status of the four wetland types, marsh wetlands recorded the highest average score of 0.756 ± 0.064, followed by offshore–coastal wetlands (0.741 ± 0.060) and river wetlands (0.710 ± 0.022). The ecological status of lake wetlands scored lowest at 0.664 ± 0.067. The relative difference between the highest and lowest scores was 13.70% (Figure 3), and the results of one-way ANOVA showed that differences between the ecological status scores of different types of wetlands were significant (F = 8.520, df = 3, $p < 0.05$). Furthermore, two-by-two comparisons revealed significant differences between lake and offshore–coastal wetlands, and between lake and marsh wetlands (lake and offshore–coastal: t = 3.550, $p < 0.05$; lake and marsh: t = 4.562, $p < 0.05$).

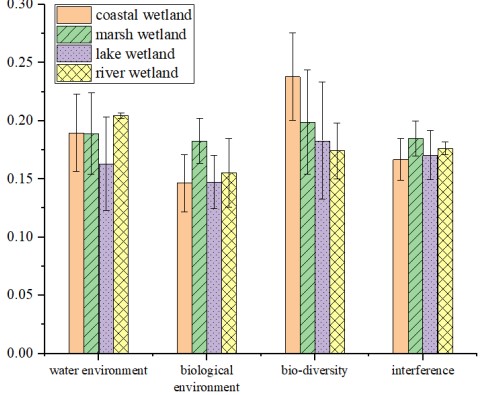

**Figure 3.** Evaluation score of different wetland types.

Comparing the scores of each type of WOII under different indicators (Figure 4), offshore–coastal wetlands scored significantly higher than other wetland types in terms of biodiversity, but scored lower than other types in terms of biological environment and disturbance pressure indicators. Marsh WOII had the highest ecological status score in terms of biological environment ($0.183 \pm 0.020$), with a score rate of 83.08%, and the ecological quality of other elements was also at a high level, with an average score rate of 76.18%. The ecological status of river wetlands scored highly in terms of water environment (average 78.67%), but had the lowest score in terms of biodiversity (average 60.05%). Lake type WOII scored poorly for all four assessment elements (average 66.77%), especially water environment (average 62.68%), which was significantly lower than for other wetland types.

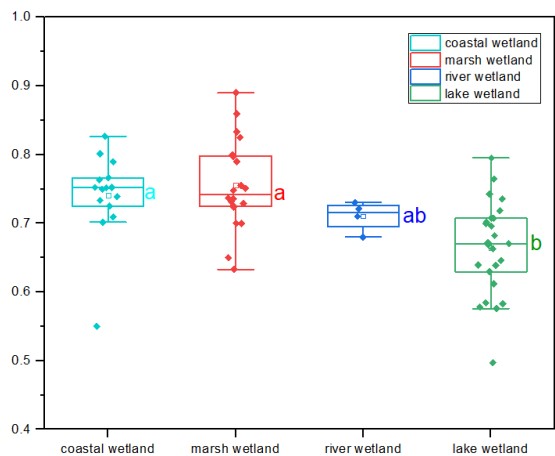

**Figure 4.** Ecological status of different wetland types.

### 3.3. Ecological Status of WOII in Different Types of Conservation Area

The establishment and management of WOII in China is carried out in nature reserves, wetland parks, and other wetland areas, and protection levels are classified as either national or provincial. According to our ecological status evaluation results, the average ecological status score of WOII in national conservation areas was $0.725 \pm 0.069$, significantly higher than that of WOII in provincial conservation areas ($0.666 \pm 0.080$; t = 2.543, $p < 0.05$) (Figure 5). Similar results were obtained by analyzing the scoring levels: the overall ecological condition of WOII in national nature reserves was good, with only 27.45% scoring 'medium' or worse, while the proportion of WOII in provincial reserves with a score of 'medium' or worse rose to 66.67%, with obvious differences.

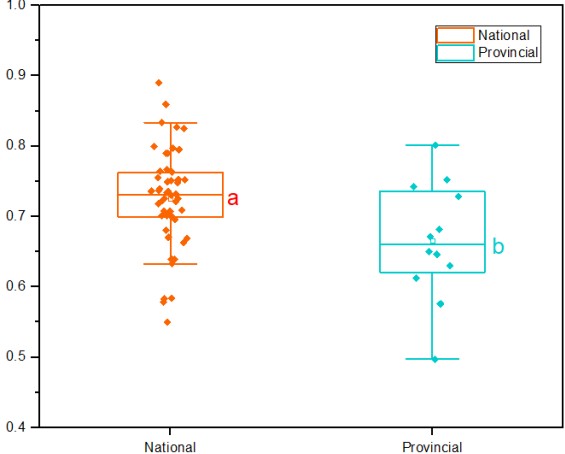

**Figure 5.** Evaluation score of different protection forms.

In terms of the different indicators, the scores of national reserve type WOII were $0.185 \pm 0.035$ and $0.164 \pm 0.027$ for water and biological environment indicators, respectively. These were significantly higher than those of provincial reserve type WOII at $0.156 \pm 0.044$ and $0.138 \pm 0.022$ (t = 3.262, $p < 0.05$; t = 2.910, $p < 0.05$), with higher percentages of 19.12% and 18.35% (Figure 6), respectively. There were no significant differences in biodiversity or disturbance pressure indicators.

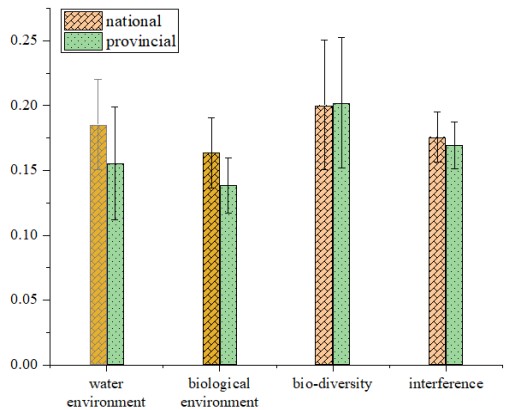

**Figure 6.** Ecological status in different protection forms.

## 4. Discussion

### 4.1. Trends in the Evolution of WOII in China

Regular monitoring and assessment can clarify the status and function of wetland ecosystems in a timely manner, and allow targeted measures to be taken to mitigate the effects of climate change, anthropogenic disturbances, and other changes in environmental conditions on the stability of wetland ecosystems. This is essential for the effective management of wetlands [29]. In 2009 and 2013, the State Forestry Administration of China completed two evaluations of the ecological status of WOII, and the proportion of WOII rated as 'excellent' was >80% [30], higher than that which was found in this study. In 2009 and 2013, the ecological status of WOII at the national level was evaluated according to three different types: endangered species, offshore–coastal type, and inland wetland type. The score assignment criteria were referred to general wetlands, and there were fewer criteria, helping to explain the differences between those results and the results herein. The ecological status of wetlands varies according to the evaluation indices and assignment criteria used. Hence, evaluation results may differ.

Our results show that the overall ecological status of WOII in China remains good, but there is obvious spatial heterogeneity and differences in ecological status among different regions and wetland types. This is consistent with the findings of other recent studies [31]. Wetlands of international importance having 'excellent' and 'good' ecological status are mainly located in northeast China, and on the north and east coasts of China, and these are mostly marsh and offshore–coastal wetlands. Wetlands of international importance having 'medium' ecological status are located on the southeast coast of China, and in the middle and upper reaches of the Yangtze and Yellow rivers, and these are mostly riverine and offshore–coastal wetlands. Wetlands of international importance having 'poor' ecological status are located in southwest China and the middle reaches of the Yangtze River, and these are mostly lake wetlands.

Overall, marsh-type WOII have a large area, high vegetation cover, rich biodiversity, and are far away from towns and cities, thus less affected by anthropogenic influences. Offshore–coastal wetlands have abundant mudflat resources and are important habitats for migratory birds, and the results of Duarte (2015) also show that the ecological status and service value of these two types of wetlands are high. Meanwhile, Wang (2021) and Xue (2018) showed through their research on the effectiveness of wetland conservation in the Yangtze River basin that important lake and river wetlands often lie adjacent to

densely populated areas, and suffer a certain degree of anthropogenic interference [32,33]. Long-term unreasonable use, such as polder occupation, has caused the degradation of lakeshore and mudflat wetland ecosystems, and the carrying capacity for wildlife is now insufficient. Hence, the ecological condition is relatively poor.

Wetlands of international importance are selected to be established in areas of high ecological quality, then maintained through a series of management and conservation measures, so that their ecological status remains higher than that of surrounding general wetlands. Different forms of protection and management intensity have influenced the ecological status of WOII distributed across China. Overall, WOII such as Zhalong and Ruoergai, which fall under national nature reserves, have management rules and regulations, management organizations and technicians, are managed with higher intensity, and generally have better ecological conditions. In contrast, WOII established in provincial nature reserves and wetland parks are relatively under-resourced and have not received sufficient attention. Some focus on ecological experiences and sightseeing, with limited ecological carrying capacity and low ecological status evaluation scores [34]. Wu (2021) compared the ecological status of wetlands in different nature reserves and concluded that nature reserves play a significant role in protecting the ecological status of wetlands, leading to high protection levels, relatively low ecological risks, and relatively good ecological status protection [20].

### *4.2. Constraints*

The ecological status of WOII remains affected by climate change, human activities, and invasive alien species, and the contradiction between ecological conservation and economic development is still prevalent. Climatic factors cause changes in the water environment of wetlands. These have a greater impact on WOII in arid areas, swamps, riverine areas, and offshore–coastal areas. Hu (2017) found that the rating of water security in wetlands showed a significant positive correlation with the ecological condition of wetlands when he studied the conservation effectiveness of important wetlands on the Sanjiang Plain [35]. In regions with relatively stable temperature and precipitation, changes in the ecological condition of WOII are often associated with human activities. Excessive human activities cause damage to the biological environment of WOII, increasing their disturbance pressure and reducing their ecological condition. Mao (2018) and Liang (2021) showed that northeast China has a concentrated distribution of important marsh wetlands. Despite increasing protection in recent years, agricultural development occupancy remains the main factor affecting WOII in areas such as the Songnun and Sanjiang plains [36,37].

The ecological condition of WOII in the middle reaches of the Yangtze River, such as Dongtinghu and Poyanghu, is greatly affected by urban development occupancy, construction, and traffic. Other parcels of land continuously encroach, and all these WOII show a trend of decreasing in area over time. In addition, WOII in southwest China are particularly threatened by pollution. After treatment in recent years, the ecological management of four important wetlands in Yunnan Province (Lashihai, Napahai, Dabaoshan, and Bitahai) has improved, but is still relatively poor compared with the Ruoergai and Yellow River Shouqu, and other WOII at the source of rivers. The factors affecting the ecological status of WOII in coastal areas mainly arise from infrastructure occupation and biological invasion. Exotic species, such as Mutual Rice Grass, have proliferated and occupied the ecological position of the original biological community, resulting in a loss of diversity of habitats for waterfowl feeding and roosting, which in turn has reduced the biodiversity and ecological status of these WOII.

### *4.3. Recommendations*

In the 30 years since accession to the Wetlands Convention, WOII have played a positive role in wetland conservation in China, but owing to the lack of management mechanisms and inadequate awareness, the management of internationally important wetland conservation needs to be further improved [38]. Meanwhile, the global development of

wetland conservation has transitioned through stages of concept determination, degradation confirmation, restoration technology, and effect evaluation. Currently, conservation efforts are shifting to wetland restoration in terms of wetland ecosystem service function assessment and response to joint anthropogenic–climate change disturbance [39]. In China, with the introduction of the wetland protection law, wetland protection and management has entered a stage of high-quality development. Strengthening the supervision and management of WOII, improving the ecological and environmental quality of WOII, and promoting the transformation of the value of ecological products of WOII under a hierarchical wetland management system have become the main directions for the future construction of WOII. These measures can also be applied in other countries in the world.

To improve the ecological quality of WOII, Wetlands International relies on the management system http://www.wetlands.org (accessed on 1 October 2022) to organize data from WOII across the globe, and regularly conducts self-assessments of WOII to understand the changing ecological quality status and trends of such wetlands. All countries should strengthen wetland monitoring, management and planning, to timely grasp the ecological status of wetlands. dynamics studies of wetlands. Furthermore, the another major tasks of wetland include the basically research of the wetland protection, the key technique research of wetland restoration, the transformation and application of the achievements in the studies of the wetland value.

The next steps are to improve the management mechanism for the protection of WOII and strengthen the legislation on WOII. Globally, there are two protection system models for WOII: one is to enact special laws for the protection of WOII, and the other is to manage WOII as protected areas [40]. At present, China has not implemented special management measures for WOII, and there are unclear responsibilities and inadequate management of WOII, which directly affects their ecological status. It is a necessary requirement for China to fulfill international conventions and assume international responsibility by formulating management measures for important wetlands, providing for the protection and restoration, monitoring and early warning, and controlled use of important wetlands at the national level, and leading the legalization of important wetlands management at the national level.

China must strengthen its management capacity by building grassroots management units of WOII, establishing a collaborative and informative notification mechanism for the protection of WOII, and forming a joint effort with populace, for wetland protection. For WOII that are in good ecological condition, China must continue to strengthen construction and management, so that these WOII have perfect protection and management facilities. For WOII that currently have a medium ecological condition, China must increase wetland management and protection, pollutant management, and reduce interfering anthropogenic activities. For WOII that are currently in poor ecological condition, China must actively carry out ecological restoration, and promote the improvement of their ecological condition. At the same time, there should be a focus on promoting the ecological value of wetland conversion [41].

## 5. Conclusions

Wetlands of international importance in China have rich biodiversity, high levels of protection, score highly on indicators of biological environment and disturbance pressure, and have an overall good ecological quality. There are differences between different regions, as the ecological condition of WOII in northeastern and northern coastal areas is better than those in other areas owing to their larger area and farther distance from cities, while WOII located in southwestern China and the middle and lower reaches of the Yangtze River remain disturbed by certain anthropogenic activities and their ecological condition requires improvement. A higher level of management and care has a positive effect on the ecological and environmental quality of WOII.

**Author Contributions:** Conceptualization, X.Z. and Y.Y.; methodology, Z.N.; formal analysis, J.Y.; writing—original draft preparation, X.Z. and Z.N. All authors have read and agreed to the published version of the manuscript.

**Funding:** This research received no external funding.

**Institutional Review Board Statement:** Not applicable.

**Informed Consent Statement:** Not applicable.

**Data Availability Statement:** Data available within the article.

**Conflicts of Interest:** The authors declare no conflict of interest.

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
