# Peer review of "Evaluation of the Ecological Status of Wetlands of International Importance in China"

_sustainability, doi:10.3390/su15043701_

Round 1

Reviewer 1 Report

Dear Authors,

Please find attached the manuscript and my comments.

I highly recommend to solve the issues I have indicated in order to see the paper published.

Regards,

Author Response

Thank you very much for your great efforts for dealing with our manuscript titled 'Evaluation of the Ecological Status of Wetlands of International 2 Importance in China' (sustainability-2172908), and we greatly appreciate the reviewers and editors for their constructive comments and suggestions. These comments and suggestions are very valuable and helpful for improving the quality of our paper. We have considered all the comments and suggestions carefully and made revisions of each point. These revisions are highlighted in red in our revised manuscript. Our responses to the comments and suggestions as well as detailed changes in the context are summarized as follows.

Reviewer:

line 16 Before mentioning the "results", you should indicate the Methods. ...and research question.

line 26 please indicate : methods, and questions driving this paper.

line 35 The References need to be "active". I mean; it is not easy to verify any references!

line 83 what is the research question?

line 89 Map to be added!

line 127 please, show the Table right here!

line 358 are those Public Policy recommendations? please add some, and also, let us know if the Methods can be reproduced somewhere else in China & the world.

line 411 missing information!

line 525 please, place these figures (and Tables) "in" text!

Response:

Point 1: line 16 Before mentioning the "results", you should indicate the Methods ...and research question.

Response 1: According to the logical relationship of importance, problems, methods, results, conclusions and suggestions, we have revised the article summary. The revised summary supplement the content of ' Methods ...and research question', and becomes more complete. (line 11-17).

Point 2: line 26 please indicate : methods, and questions driving this paper.

Response 2: Referring to the first suggestion reply, we have revised the corresponding contents of the abstract and article.

Point 3: The References need to be "active". I mean; it is not easy to verify any references!

Response 3: Thank you for your suggestion. We sorted out and confirmed the availability of the manuscript references again, and revised the problematic parts. The references in this paper are 'active'. (line 471-473, line480-485, line523-524, line531-534)

Point 4: line 83 what is the research question?

Response 4: The core of academic papers is to bring out questions and solve them through research and analysis. According to experts' suggestions, we supplemented the research questions in the manuscript. (line 86-91).

Point5: line 89 Map to be added!

Response 5: We move the attached figure into the text. Other attached figures are also adjusted. (line 103, line249, line264, line286).

Point 6: line 127 please, show the Table right here!

Response 6: We move the attached table into the text. Other attached tables are also adjusted. (line 145, line174).

Point7: line 358 are those Public Policy recommendations? please add some, and also, let us know if the Methods can be reproduced somewhere else in China & the world.

Response 7: This part is indeed a proposal for the management of WOII, focusing on the improvement of wetland ecological quality in China. According to the latest trend of wetland protection in the world, relevant contents are supplemented, and those methods can be reproduced somewhere else in China & the world.(line 378-397)

Point8: line 411 missing information!

Response 8: We have supplemented relevant information (line 431-438).

Point 9: line 525 please, place these figures (and Tables) "in" text!

Response 9: We move the attached tables and figures into the text.

Besides the revisions of each point from the reviewers, we corrected some format mistakes. Any changes are highlighted in the revised manuscript.

We would like to express our great thanks to you and the reviewers for the comments and suggestions on our paper.

Looking forward to hearing from you.

Thank you and best regards.

Reviewer 2 Report

The paper is well written, but there are many very long paragraphs that need to be broken up into shorter paragraphs. This is a common problem with authors from China, so I assume that it is due to the different ways in which the languages are constructed grammatically. You might try applying the simple rule that: if a paragraph is longer than half a page, it should be broken up into 2-3 shorter paragraphs.

A principle of scientific writing is that a reader should be able to repeat the methodology used, and also review the results found. Neither is possible with this paper.

·         How were the index values assigned. No detail is given. A reader cannot apply the same methodology – e.g. to more parks/reserves in China, or anywhere else

·         A Table needs to be included that gives the raw index values for each of the 63 parks/reserves. Otherwise the paper cannot be linked to other similar analyses. The only data provided are in summary form, and the raw data need to be available (they could be provided under “additional materials”).  

Minor detail changes

Line 96-99. Sentence does not make sense.

Line 142. Brackets around Fi*Wi to ensure clarity of formula.

i.e. Σ(Fi*Wi)

Line 397-399. Sentence does not make sense.

Statistical analysis seems appropriate.

Author Response

Thank you very much for your great efforts for dealing with our manuscript titled 'Evaluation of the Ecological Status of Wetlands of International 2 Importance in China' (sustainability-2172908), and we greatly appreciate the reviewers and editors for their constructive comments and suggestions. These comments and suggestions are very valuable and helpful for improving the quality of our paper. We have considered all the comments and suggestions carefully and made revisions of each point. These revisions are highlighted in red in our revised manuscript. Our responses to the comments and suggestions as well as detailed changes in the context are summarized as follows.

Reviewer:

The paper is well written, but there are many very long paragraphs that need to be broken up into shorter paragraphs. This is a common problem with authors from China, so I assume that it is due to the different ways in which the languages are constructed grammatically. You might try applying the simple rule that: if a paragraph is longer than half a page, it should be broken up into 2-3 shorter paragraphs.

A principle of scientific writing is that a reader should be able to repeat the methodology used, and also review the results found. Neither is possible with this paper.

How were the index values assigned. No detail is given. A reader cannot apply the same methodology – e.g. to more parks/reserves in China, or anywhere else. A Table needs to be included that gives the raw index values for each of the 63 parks/reserves. Otherwise the paper cannot be linked to other similar analyses. The only data provided are in summary form, and the raw data need to be available (they could be provided under “additional materials”). 

Minor detail changes

Line 96-99. Sentence does not make sense.

Line 142. Brackets around Fi*Wi to ensure clarity of formula. i.e. Σ(Fi*Wi)

Line 397-399. Sentence does not make sense.

Statistical analysis seems appropriate.

Response:

Point 1: There are many very long paragraphs that need to be broken up into shorter paragraphs. You might try applying the simple rule that: if a paragraph is longer than half a page, it should be broken up into 2-3 shorter paragraphs.

Response 1: Thank you for your comments. We sorted out the paragraphs of the article and re-divided the long paragraphs to better highlight the focus of each paragraph(line 57, line 189, line 361).

Point 2: How were the index values assigned. No detail is given. A reader cannot apply the same methodology – e.g. to more parks/reserves in China, or anywhere else. A Table needs to be included that gives the raw index values for each of the 63 parks/reserves. Otherwise the paper cannot be linked to other similar analyses. The only data provided are in summary form, and the raw data need to be available (they could be provided under “additional materials”).

Response 2: We have revised the evaluation method to be more specific in the revised manuscript. In the process of designing the evaluation form, according to the relevant standards(Wetland Ecological Quality Assessment Specification) and the opinions of experts in the field, we used the evaluation matrix results to assign the weight. (line 120-125).

According to expert suggestions, we have listed the table of raw index values for each of the 63 international importance wetlands as 'additional materials' in the attachment table, to facilitate comparative analysis by other researchers.

Minor edits are needed as follows:

Point1: Line 96-99 Sentence does not make sense.

Response 1: The intention of this sentence is to illustrate varies greatly between different important wetlands and pave the way for their ecological quality analysis. According to the expert opinions, we modified the sentence to remove the description of maximum and minimum area, and retained the description of wetland rate index. line (100-102).

Point 2: Line 96-99 Sentence does not make sense.

Response 2: We have revised it to 'Σ(Fi*Wi)' (line 152).

Point3: Line 397-399. Sentence does not make sense.

Response 3: The intention of this sentence is to illustrate the results of 'promoting the ecological value of wetland conversion'. According to expert suggestions, it has been deleted in the revised manuscript.

Besides the revisions of each point from the reviewers, we corrected some format mistakes. Any changes are highlighted in the revised manuscript.

We would like to express our great thanks to you and the reviewers for the comments and suggestions on our paper.

Looking forward to hearing from you.

Thank you and best regards.

Yours sincerely,

Xinsheng Zhu

Name: Ziru Niu

Round 2

Reviewer 1 Report

The authors have significantly improved their manuscript. They corrected errors and answered all my questions. The manuscript is ready for publication.